# Spatial coupling relationship between older adults and elderly care resources in the Yangtze River Delta

**Lianxia Wu[1,2], Linyi Qian[3], Yinhuan Li[3]\*, Zuyu Huang[4], Weihua Guan[5,6]**

1 Population Research Institute, School of Social Development, East China Normal University, Shanghai, China, 2 Aging Research Base of China National Committee on Aging (East China Normal University), Shanghai, China, 3 Key Laboratory of Advanced Theory and Application in Statistics and Data Science-MOE, School of Statistics, East China Normal University, Shanghai, China, 4 School of Public Administration, Hunan University, Changsha, Hunan, China, 5 School of Geography, Nanjing Normal University, Nanjing, China, 6 Collaborative Innovation Center for Development and Utilization of Geographic Information Resources in Jiangsu Province, Nanjing, China

* 52204404004@stu.ecnu.edu.cn

**Data Availability Statement:** The seventh National Population Census and local yearbooks data are available from http://www.stats.gov.cn/english/.

## Abstract

The imbalance between supply and demand of elderly care resources in the Yangtze River Delta is increasing. By the older adult agglomeration, spatial cluster analysis, hotspot analysis, and coupling coordination model, this study explores the spatial coupling relationship between older adults and elderly care resources in the Yangtze River Delta in 2020 from the perspective of a supply-and-demand balance. The results demonstrate that: (1) population aging is mainly in the moderate aging stage, followed by the primary aging stage; (2) there are significant spatial differences in elderly care resources on the urban scale in the Yangtze River Delta; and (3) elderly care resources and the older adults in the Yangtze River Delta are mostly highly coupled. However, Nantong, with the highest degree of aging, has a serious mismatch in life service resources and ecological environment resources. The social security resources and medical resources of provincial capital cities with low aging are mismatched. Medical and health resources in underdeveloped areas are seriously mismatched. The social security resources are barely matched in Shanghai. A path for optimizing the spatial allocation of elderly care resources is proposed. This research offers a decision-making reference for coordinating elderly care resources distribution.

## Introduction

With increased life expectancy and a sustained decline in fertility rates, population aging has become a primary trend in population development. As the World Population Prospects 2022 illustrates, the proportion of the worldwide population aged 65 and above is estimated to increase from 10% in 2022 to 16% in 2050, while the global number of older adults aged 65 and above is about to become more than twice the number of children under the age of 5 and will reach the number of children under the age of 12 by 2050 [1]. According to China's Seventh National Population Census, the older adults aged 60 and above (excluding

Earth Big Data are available from https://earthbigdata.com/.

**Funding:** National Social Science Fund Youth Project in China (19CRK010), Fundamental Research Funds for the Central Universities (2020ECNU-HLYT048, 2022QKT001), the Humanity and Social Sciences Foundation of Ministry of Education of China (19YJC840032, 21YJA840001), National Natural Science Foundation (12171158, 42001161), the State Key Program of National Natural Science Foundation of China (71931004), Nanjing Social Science Foundation Project(23YB02). National Social Science Fund Youth Project in China (19CRK010) supported the fund leader in designing this study and deciding on the publication of the paper. Fundamental Research Funds for the Central Universities (2020ECNU-HLYT048) supported authors in writing and polishing the manuscript. Fundamental Research Funds for the Central Universities (2022QKT001), National Natural Science Foundation (12171158), and the State Key Program of National Natural Science Foundation of China (71931004) offered the financial resource of softwares. The Humanity and Social Sciences Foundation of Ministry of Education of China (19YJC840032) and Nanjing Social Science Foundation Project(23YB02) played a role in collecting, cleaning and processing data. The Humanity and Social Sciences Foundation of Ministry of Education of China (21YJA840001) and National Natural Science Foundation (42001161) played a role in supporting the project administration and materials collection.

**Competing interests:** The authors have declared that no competing interests exist.

residents of Hong Kong, Macao, and Taiwan) reached 264.02 million in 2020, and accounted for 18.7% of China's total population, with 13.5% (190.64 million) aged 65 and above [2]. Data from the National Health Commission of the People's Republic of China demonstrates that by the end of 2021, the proportion of older people over 60 increased to 18.9% of the total population. Furthermore, the number of people aged 65 and above exceeded 200 million, reaching 200.56 million (14.2%), while the dependency rate of those aged 65 and above dramatically increased from 19.7% in 2020 to 20.8% in 2021 [3, 4]. Population aging in China is accelerating at an unprecedented speed and scale, and the country is expected to become a super-aging society by 2033 [5] and to have the largest older adults in the world by 2070. This trend indicates that the Chinese government faces the problem of how best to support the older adults.

As a pioneering region in China's economic and social development, the Yangtze River Delta (including Shanghai, Jiangsu, Zhejiang, and Anhui, 41 cities, Fig 1) was one of the earliest China's regions to experience the population transition and aging-related problems. According to the Seventh National Population Census, in 2020, the Yangtze River Delta had a permanent resident population of 230 million, including 35.5 million older adults aged 65 and above, accounting for approximately 15.1% of the total population. The degree of population aging in the Yangtze River Delta is significantly higher than that in the whole country. An increasingly prominent issue is the allocation of resources for elderly care. Affected by many factors such as regional economic development level, resource endowment differences, and changes in population, family structure, and a culture of filial piety, the older adults in the Yangtze River Delta faces multifarious elderly care issues and care service circumstances.

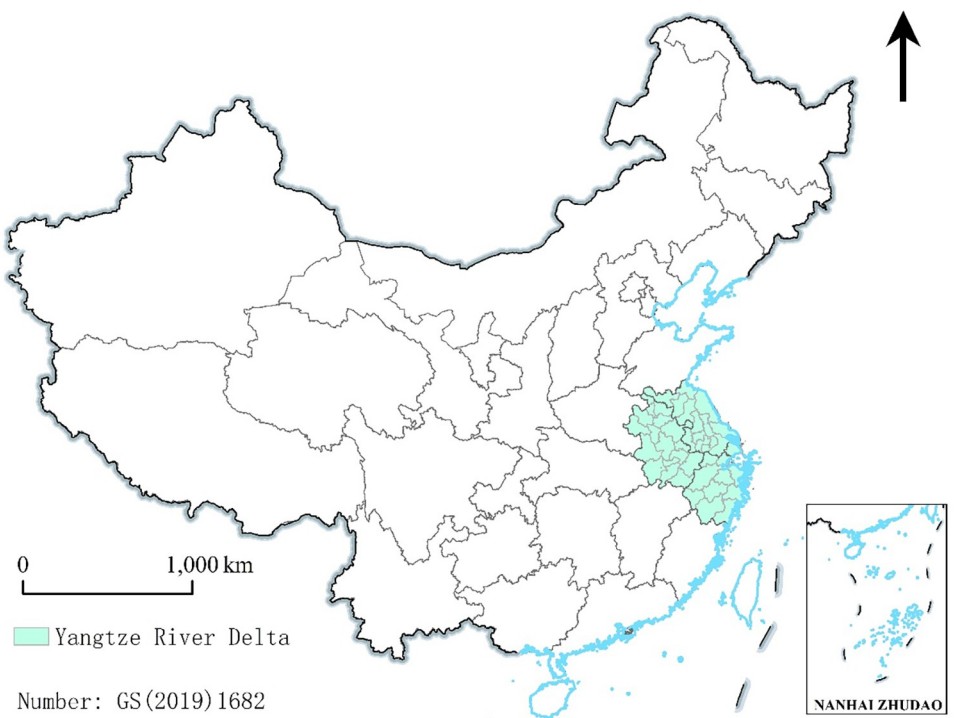

**Fig 1. The map of the Yangtze River Delta in China.**

## Literature review

Research on social elderly care services can date to 1933. Ogburn [6] suggested that the functionality of home-based elderly care was weakening, and more research was required on social elderly care services. In 1984, Evashwick et al. [7] believed it important to efficiently plan and develop health services for older adults to balance the demand for services with the available resources. The strategic allocation of elderly care resources is vital for the development of social elderly care services, affected by factors like the economy and government policies [8, 9]. National conditions and cultural differences have led to the widespread implementation of pension systems in foreign countries faced with the problem of population aging. In most Western countries, the institution-based elderly care was developed earlier, and the related management and evaluation systems, laws, and regulations are relatively complete. Therefore, elderly care institutions can operate in a good market environment [10]. Comparatively speaking, the development of Chinese elderly care institutions is later, their management system is not well-established, and the allocation of elderly care resources is not sufficiently rational, requiring urgent attention.

With the development of social elderly care services in recent years, many domestic scholars have presented concepts or classifications of elderly care resources. Zhao [11] defined elderly care resources as all resources available to older adults to cope with elderly care issues in the context of urban elderly care. Such resources are material elements that offer various services to older adults, including human, material, and financial resources [12]. Broadly, we believe that they can be defined as all accessible resources that meet the demands of older adults. From this perspective, elderly care resources are not limited to urban elderly care but include those needed in rural areas. More specifically, elderly care resources comprise not only social, material, and tangible resources (such as the economy and environment) but also family, spiritual, and intangible resources (such as the "time bank" retirement mode). In a narrow sense, which is the perspective we adopt in this study, elderly care resources mainly constitute all available resources needed for social elderly care. In this study, it refers to material and tangible resources that are accessible to older adults.

Some scholars summarized elderly care services as the daily life care and professional health care, including nursing, education, culture and entertainment [13]. Ji and Wells [14] divided elderly social support resources into five categories: external financial resources, medical resources, physical resources, family resources, and self-resources. Based on the above literature and our research scope, we divide elderly care resources into four aspects: life service resources, ecological environment resources, social security resources and medical and health resources. Life service resources are various services provided to meet the daily needs of older adults [15], such as institution and community elderly care facilities. Ecological environment resources refer to various natural resources related to the life and health of older adults, such as the park green space and air temperature. Social security resources are a series of welfare measures taken by governments or social organizations to support older adults' lives, such as pension insurance. Medical and health resources refer to the various assets, facilities, personnel, and materials that are available to support healthcare and medical services needed by older adults within a healthcare community or a hospital, such as the number of hospital beds and doctors.

Social elderly care is potentially a high-quality way of supporting older adults [16]. As the population ages rapidly, social elderly care resources are insufficient, and social care development for older adults is imbalanced in different regions [17, 18]. The number of elderly care institutions and hospital beds varies from different provinces, and it is difficult for elderly care institutions to satisfy their needs [19]. There are not only interprovincial differences in social

elderly care resources but also regional differences among the east, central, and west. The unbalanced and inadequate allocation of elderly care facilities remains a limitation on developing quality facilities, particularly in large and medium-sized cities [20]. Consequently, many scholars have gradually shifted their attention to the elderly care resource allocation in communities, institutions, cities, and rural areas and offered corresponding optimization suggestions. For example, Xu and Liu [21] optimized the allocation of resources using a dynamic monitoring method for integrated medical and elderly care. To explore whether spatial allocation is balanced, many studies have focused on the accessibility of elderly care resources in several regions, such as Nanjing and downtown Shanghai [22–25], commonly measured by the 2-step floating catchment area (2SFCA) method [26–28] and limited in a spatial scope.

In addition, substantial research has been conducted on the relationship between supply and demand for elderly care resources (such as healthcare services for older adults) [12, 29], which is an essential index to reflect the allocation of elderly care resources. Coupling theory is one of significant tools to explore the coupling between supply and demand and can provide scientific perspectives to reasonably regulate supply and demand [30]. Based on the coupling coordination model, scholars have investigated the spatial-temporal evolution and the development trends of the supply and demand of elderly care resources, with results that the coupling degree is barely imbalanced and close to imbalanced in China, and regional differences between supply and demand are obvious [31]. By analyzing the degree of coupling, the supply and demand situation for elderly care resources can be established, thus contributing to further optimizing the supply of these resources and better satisfying the needs of older adults in various regions [32]. From the perspective of existing research, the allocation of social resources for older adults has become a hot issue closely related to population aging. Research on the differences in their distribution is important as it is a key aspect of social elderly care. A research gap exists regarding the allocation of and differences in elderly care resources between or within different regions. Given the increasing prominence of social elderly care in China's elderly care service system, the rational allocation of elderly care institutions and medical beds is a significant factor in promoting the equalization of elderly care services in China and coping with population aging.

Overall, research priorities and contexts worldwide regarding the distribution of older adults and the allocation of elderly care resources have become clear, and previous studies of the spatial allocation and optimization of elderly care resources have provided a solid theoretical foundation for this study. However, these investigations are limited in various ways. In terms of the research content, many were conducted on elderly care facilities, medical and health care, and pension systems to assess the relationship between supply and demand, while few focused on coupling coordination development and the optimal allocation of elderly care resources as a whole, thus resulting in a lack of theoretical support. However, the problem of supporting older adults is associated with all aspects of elderly care resources, meaning that integrating all resources, studying their spatial allocation, and optimizing elderly care resources are required.

The concerning research methods lack the technical support and cross-disciplinary approach to analysis. Nevertheless, studying the distribution of older adults and the allocation of elderly care resources requires such an interdisciplinary and cross-sectional approach. With the development of human society, the distribution of elderly care resources remains uneven, while the older adults continue to grow. There is a desperate need to eliminate disciplinary barriers and realize cross-disciplinary cooperation such as between management, statistics, and information technology to cope with population aging and promote the sustainable development of elderly care.

Regarding the research scope, the existing studies on the allocation of elderly care services have mainly focused on the whole country or a city (especially a central city or even a downtown area), while few concentrated on the allocation of elderly care resources in representative regions. Issues such as local protectionism and other administrative barriers mean that policy implementation is not well served when the research scope is too extensive or too narrow. In the new period when the integration of the Yangtze River Delta has become a national strategy, it is paramount to promote the integration of elderly care resources. Improving spatial allocation to the best possible extent is a feasible solution to address the real problems of supporting older adults in China and would have important practical implications. What are the characteristics of the distribution of older adults and care service resources in each city? Is there a coupling relationship between them on the supply of and demand for elderly care resources? How can the spatial allocation of support resources for older adults be further optimized? This research was conducted to address these points.

In terms of content, this paper researches the coupling and coordinated development of elderly care resources as a whole, integrates all elderly care resources, and explores the path to optimize the spatial allocation of elderly care resources. It can enrich the theoretical connotations of balanced development, sustainable development, resource sharing, and regional integration of elderly care resources, laying a theoretical foundation for the development of gerontology. In terms of research theory and methods, this study integrates interdisciplinary theories and methods such as geography, statistics, and management, which is conducive to eliminating disciplinary barriers and promoting the development of interdisciplinary theories and methods. In terms of research scope and countermeasures, this study utilizes the integrated national strategy of the Yangtze River Delta, selects national key regions such as the Yangtze River Delta as the research object, and proposes that optimizing elderly care resources in the Yangtze River Delta based on the integration of the Yangtze River Delta is conducive to breaking through local protectionism and administrative barriers, and transitioning from "regionalization" and "fragmentation" governance to "integration" and "integrity" elderly care social governance models.

This study considered 41 cities in three provinces and one municipality in the Yangtze River Delta (Fig 1). It adopted relevant techniques such as geographic information systems and integrated multidisciplinary theories of geography, demography, and management. In addition, based on supply and demand theory, coupling coordination development theory, and integration theory, the spatial distribution and coupling allocation relationship between the older adults (65 years and above, the same hereinafter) and elderly care resources in the cities of this region were investigated. An optimal configuration path for elderly care resources is proposed in the integration process of the Yangtze River Delta and provides empirical evidence for other key regions and the entire country to appropriately arrange the distribution of resources, evenly develop the elderly care, and effectively utilize elderly care resources. Moreover, the analysis provides a scientific basis to adjust measures to local conditions, implement specific policies, break down administrative divisions, and promote the integral development of regional elderly care.

## Research methods and data sources

### Spatial cluster analysis, hotspot analysis

Using GIS software, spatial clustering analysis and hotspot analysis were conducted to explore the spatial patterns of the older adults and elderly care resources in the Yangtze River Delta. Hotspot analysis refers to the adoption of the Getis-ord $G_i^*$ index to identify hot spots (high-value clusters) and cold spots (low-value clusters) in the research region and explore the spatial

pattern of the research object through the spatial distribution of cold and hot spots. The equation is as follows [33, 34]:

$$G_i^* = \frac{\sum_{j=1}^{n} W_{ij} x_j}{\sum_{j=1}^{n} x_j}$$

$$j \neq i$$

(1)

For comparative purposes, $G_i^*$ is standardized by

$$Z(G_i^*) = \frac{G_i^* - E(G_i^*)}{\sqrt{Var(G_i^*)}}$$

(2)

where $E(G_i^*)$ and $Var(G_i^*)$ represent the expectation and variance of $G_i^*$ respectively, and $W_{ij}$ is the spatial weight between $i$ and $j$. When the spatial range is adjacent, the weight is 1; otherwise, it is 0. If $Z(G_i^*) > 0$ and the hypothesis test is statistically significant, the higher value means the more intense clustering around $i$ (hot spot). However, $Z(G_i^*) < 0$ and a significant hypothesis test indicate low-value clustering of $i$ (cold spot).

## The aggregation degree of the older adults

The aggregation degree of the older adults refers to the degree of aggregation in a region relative to the whole country; that is, the proportion of the older adults clustered in a region accounting for 1% of the national land area [35]. The expression is given as follows:

$$JJD_i = \frac{\left(\frac{P_i}{P_n}\right) * 100\%}{\left(\frac{A_i}{A_n}\right) * 100\%} = \frac{\frac{P_i}{A_i}}{\frac{P_n}{A_n}},$$

(3)

where $JJD_i$ is the degree of concentration of the older adults in region i, $P_i$ is the scale of the older adults in region i, $A_i$ is the land area of region i, $P_n$ is the scale of the national older adults, and $A_n$ is the land area of the entire country.

## Coupling degree model and coupling coordination model

Coupling degree model and indicator system

The grey relation analysis is to measure the intensity and order of the relation between various factors based on the grey relation grade and to quantitatively analyze the dynamic development process of a system. Compared with other methods, it can more accurately explain the degree of closeness and spatial regularity among factors, and it is superior to the classical mathematical methods when dealing with data with unclear connotations and extensions.

This study employed the grey relation analysis to construct a coupling degree model, comprehensively evaluated the coupling relationship between older adults and elderly care resources, and wholly revealed the spatial configuration patterns of the supply of and demand for the elderly care in the Yangtze River Delta. The procedures to construct the coupling degree model between the older adults and elderly care resources are as follows.

Firstly, sequences of the older adults ($X_i$, *aged* 65 *and above*) and elderly care resources ($Y_i$) are determined. The sequence of older adults consists of two indicators: aging population and dependence rate of older adults, measured by the proportion of older adults ($X_1$) and the proportion of older adults in the working-age population (15–64 years) ($X_2$), respectively. The sequence of elderly care resources has four indicators: life service resources, ecological environment resources, social security resources, and medical and health resources (see Table 1).

**Table 1. Indicators of the coupling system between older adults and elderly care resources.**

| First-level Indicators | Second-level Indicators | Third-level Indicators |
|---|---|---|
| Older adults system $X$ | Aging population | $X_1$—the proportion of older adults (65 years and above) |
| | Dependence rate of older adults | $X_2$—the proportion of older adults in the working-age population (15–64 years) |
| Elderly care resources system $Y$ | Life service resources | $Y_1$—the number of elderly care facilities per 10,000 older adults |
| | Ecological environment resources | $Y_2$—the area of park green spaces per 10,000 older adults |
| | Social security resources | $Y_3$—the number of urban basic old-age insurance participants per 10,000 older adults |
| | Medical and health resources | $Y_4$—the number of medical beds per 10,000 older adults |

Based on the principles of scientificity, feasibility, availability, and comparability [36], we select the following evaluation indicators [37–45]: the number of elderly care facilities per 10,000 older adults ($Y_1$), the area of park green spaces per 10,000 older adults ($Y_2$), the number of urban basic old-age insurance participants per 10,000 older adults ($Y_3$), and the number of medical beds per 10,000 older adults ($Y_4$), corresponding to the four second-level indicators, respectively. The more elderly care facility institutions, the more abundant life service resources are available to the older adults; the more park green spaces every older adult owns, the more friendly communities will be developed; the more participants, the more social security resources are supplied; the more medical beds, the more elderly inpatients can be adopted [31]. It should be noted that, according to the reference [22] and the principles of data accessibility, the elderly care facilities categorized under life service resources that is the first type of elderly care resources, mainly refer to the resources of social elderly care institutions in this research, which provide various services and support for the older adults. The elderly care facilities contain elderly apartments, elderly daycare institutions, nursing stations, nursing homes, medical institutions. Based on the Earth Big Data, the data of elderly care facilities can be extracted using Python.

Secondly, the range method is used to normalize the data of all indicators, to avoid inconvenient comparison due to different physical meanings of various factors in the system. Finally, the relation coefficient is calculated by Eq (4). Then, the coupling relation model and the coupling degree model are successively established to obtain the coupling relation grade and coupling degree, respectively. The related formulas are as follows:

$$R_{ij}(t) = \frac{min_i \, min_j \, |Xi'(t) - Yj'(t)| + \rho \, max_i \, max_j \, |Xi'(t) - Yj'(t)|}{|Xi'(t) - Yj'(t)| + \rho \, max_i \, max_j \, |Xi'(t) - Yj'(t)|}, \tag{4}$$

$$\gamma_{ij} = \frac{1}{k} \sum\nolimits_{i,j=1}^{k} R_{ij}(t) \qquad (k = 1, 2, \ldots, n), \tag{5}$$

$$di = \frac{1}{l} \sum\nolimits_{i=1}^{l} r_{ij} \quad (i = 1, 2, \cdots, l; j = 1, 2, \cdots, m), \tag{6}$$

$$dj = \frac{1}{m} \sum\nolimits_{j=1}^{m} r_{ij} \quad (i = 1, 2, \cdots, l; j = 1, 2, \cdots, m), \tag{7}$$

$$C(t) = \frac{1}{m \times l} \sum\nolimits_{i=1}^{l} \sum\nolimits_{j=1}^{m} R_{ij}(t). \tag{8}$$

In Eq (4), $R_{ij}(t)$ is the relation coefficient between the $i$th older adults indicator and the $j$th elderly care resources indicator of each city in the Yangtze River Delta at time $t$; $Xi'(t)$, $Yj'(t)$ are normalized values of indicators of older adults and elderly care resources at time $t$, respectively; $\rho$ denotes the distinguishing rate, reflecting the significance of difference between relation coefficients, which is generally 0.5.

In Eq (5), $\gamma_{ij}$ denotes the relation grade (between 0 and 1), and $k$ is the sample size. When $\gamma_{ij}$ = 1, it indicates that the correlation between $X_i(t)$ and $Y_j(t)$ is large, and they have completely consistent changes and very strong coupling effect. When $0<r_{ij}<1$, $X_i(t)$ and $Y_j(t)$ have the coupling correlation; the larger $\gamma_{ij}$, the stronger the coupling effect. It consists of the following 4 categories: high correlation ($0.85<r_{ij}\leq1$), which has very strong coupling effect of indicators between the two systems; relatively high correlation ($0.65<r_{ij}\leq0.85$), which has relatively high coupling effect; moderate correlation ($0.35<r_{ij}\leq0.65$), which has moderate coupling effect; low correlation(($0<r_{ij}\leq0.35$), which has weak coupling effect.

In Eqs (6)–(8), $l$ and $m$ are the numbers of indicators in the two systems, respectively. In Eq (6) and (7), $d_i$ denotes the average relation grade between the elderly care resources system and the $i$th indicator of the older adults system, while $d_j$ denotes that between the older adults system and the $j$th indicator of the elderly care resources system, and $C(t)$ in Eq (8) is the coupling degree.

(2) Coupling coordination model

However, the comprehensive evaluation of overall elderly care resources, including medical resources, elderly care facilities, social security, can only show the close relationship between the whole resources and the older adults, and cannot deeply analyze the matching degree and the coupling coordination of each resource and the older adults. To better appreciate our study, this paper utilized a coupling coordination model to investigate the spatial distribution, matching relationship, and divisions between elderly care resources (such as elderly care facilities, park green space, social security) and population aging. The equation is expressed as [46, 47]:

$$C = 2\sqrt{X \times \frac{Y}{(X + Y)^2}}, \tag{9}$$

where $C\in[0,1]$ is the coupling degree of elderly care resources and the older adults (i.e., the spatial adaptation value). The larger the value of $C$, the better the resonant coupling between them. $X$ represents the older adults, measured by the proportion of the population aged 65 and above in the total population. $Y$ represents the level of elderly care resources, i.e., life service resources, ecological environment resources, social security resources, and medical and health resources, which are respectively measured by the number of elderly care facilities, park green spaces, urban workers covered by basic pension insurance, and medical beds per ten thousand older people.

## Data sources

The primary data sources for this study were the seventh National Population Census, local statistical yearbooks, and Earth Big Data. The proportion of the older adults in each region of the Yangtze River Delta was derived from the seventh National Population Census data. Data on medical beds and social security were mainly obtained from the 2020 statistical yearbooks of various regions. According to Earth Big Data, other data (such as the number and distribution of park green spaces, and elderly care facilities) can be extracted and processed using Python.

## Spatial distribution of the older adults and elderly care resources in the Yangtze River Delta

### Spatial distribution of population aging

(1) Population aging is primarily in the moderate aging stage, followed by the primary aging stage. In particular, Nantong and Taizhou, which are at the severe aging stage, should be prioritized in allocating elderly care resources.

The aging coefficient ($W$) is a commonly used indicator to measure the degree of aging and describes the percentage of the older adults (65 years and above) in the total population in a certain region and at a certain point. Using the aging coefficient, population development patterns are roughly classified as 'young stage' ($W \leq 4\%$), 'adult stage' ($4\% < W < 7\%$), and 'aging stage' ($W \geq 7\%$) [48]. In 2020, the aging population index of the entire Yangtze River Delta region exceeded 7%. To facilitate an in-depth analysis of the characteristics of urban population aging in the Yangtze River Delta, this study combined China's national conditions and international standards (Wu et al., 2021) to divide aging into the 'primary aging stage' ($7\% \leq W < 14\%$), the 'moderate aging stage' ($14\% \leq W < 21\%$), and the 'severe aging stage' ($W \geq 21\%$) (see Fig 2). The primary aging stage contains 12 cities, mainly located in northern Anhui Province such as Fuyang, Taizhou, Suzhou, and Suqian; the southeastern coastal areas of Zhejiang Province; and some areas in Jiangsu Province. They account for 29.27% of all cities in the Yangtze River Delta. The moderate aging stage comprises 27 cities widely distributed in Yangzhou, Quzhou, Shanghai, Shaoxing, etc., thus representing 65.85% of the total number of cities studied. The severe aging stage is mainly concentrated in Nantong and Taizhou, and accounts for 4.88% of the total. From this perspective, population aging in the Yangtze River Delta region is dominated by the moderate aging stage, with the primary aging stage next in significance. Nantong and Taizhou, which experience the severe aging stage, should be especially considered when allocating elderly care resources.

(2) Hot spots such as Nantong, Yancheng, Taizhou, and Yangzhou have a greater demand for elderly care, while cold spots such as Hefei, Jinhua, and Suzhou have relatively less demand.

The Getis-ord $G_i^*$ index was used to obtain the local $G_i^*$ statistic of each city unit in 2020, which was divided into four categories by natural break classification. The spatial distribution patterns of the hot and cold spots were obtained using GIS (see Fig 3).

In the Yangtze River Delta, there are relatively distinct spatial discrepancies between the western and eastern areas in terms of the cold and hot spots of an aging population. Hot spots are clustered in the eastern areas of Jiangsu Province, such as Nantong and Yancheng, whereas cold spots are mainly distributed in Hefei, Hangzhou, and Wenzhou among others. In addition, there are many sub-cold spots in the north, such as Fuyang, and in the southwest, such as Lishui, whereas sub-hot spots are relatively concentrated in the central and western areas (such as Anqing and Wuhu) and the east (such as Huai'an), thereby indicating that the older adults in the Yangtze River Delta has a core-periphery structure. Specifically, the demand for elderly care in hot spots (such as Nantong, Yancheng, Taizhou, and Yangzhou) is relatively large, while that in cold spots (such as Suzhou, Hefei, Jinhua, and Wenzhou) is relatively small.

(3) Older adults aggregation and demand for elderly care in Shanghai are much higher than in other cities.

The coefficient of aging is a static index used to measure aging, but it is one-sided, and it is not easy to characterize relative or comprehensive aging conditions. Therefore, the degree of agglomeration of older adults was used to measure the spatial clustering of the older adults in every city of the Yangtze River Delta relative to that of the national older adult population.

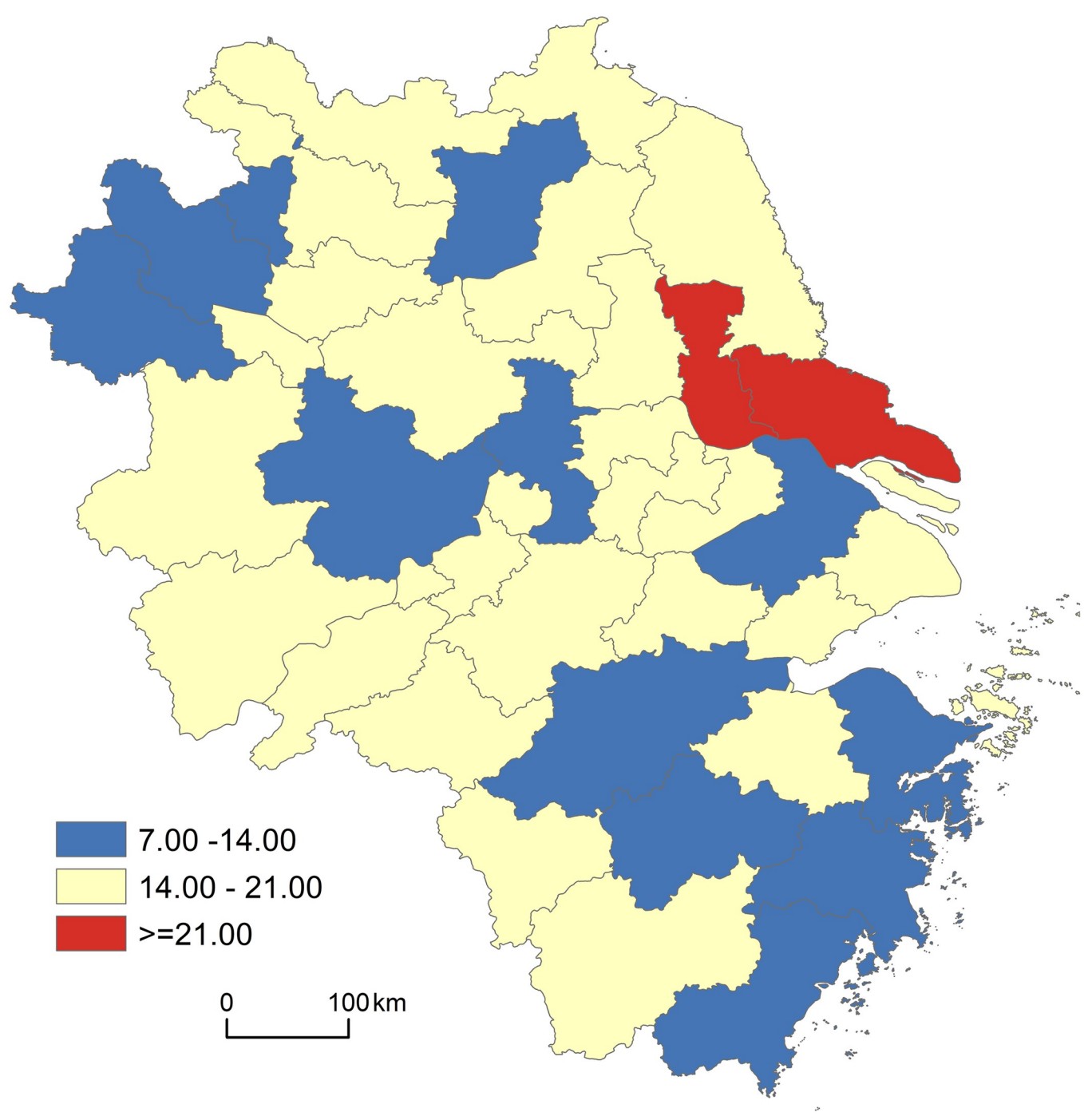

**Fig 2. Spatial distribution of population aging degree in the Yangtze River Delta.**

Using Eq (3), we calculated the degree of agglomeration of the older adults in urban areas of cities in the Yangtze River Delta in 2020. Combined with relevant scholarly classification and evaluation criteria of the national population concentration [30], the older adults in the Yangtze River Delta can be divided into five types: super-high-density area (JJD>12), high-density area (9<JJD≤12), medium-density area (6<JJD≤9), low-density area (3<JJD≤6) and sparse area (JJD≤3) (see Fig 4).

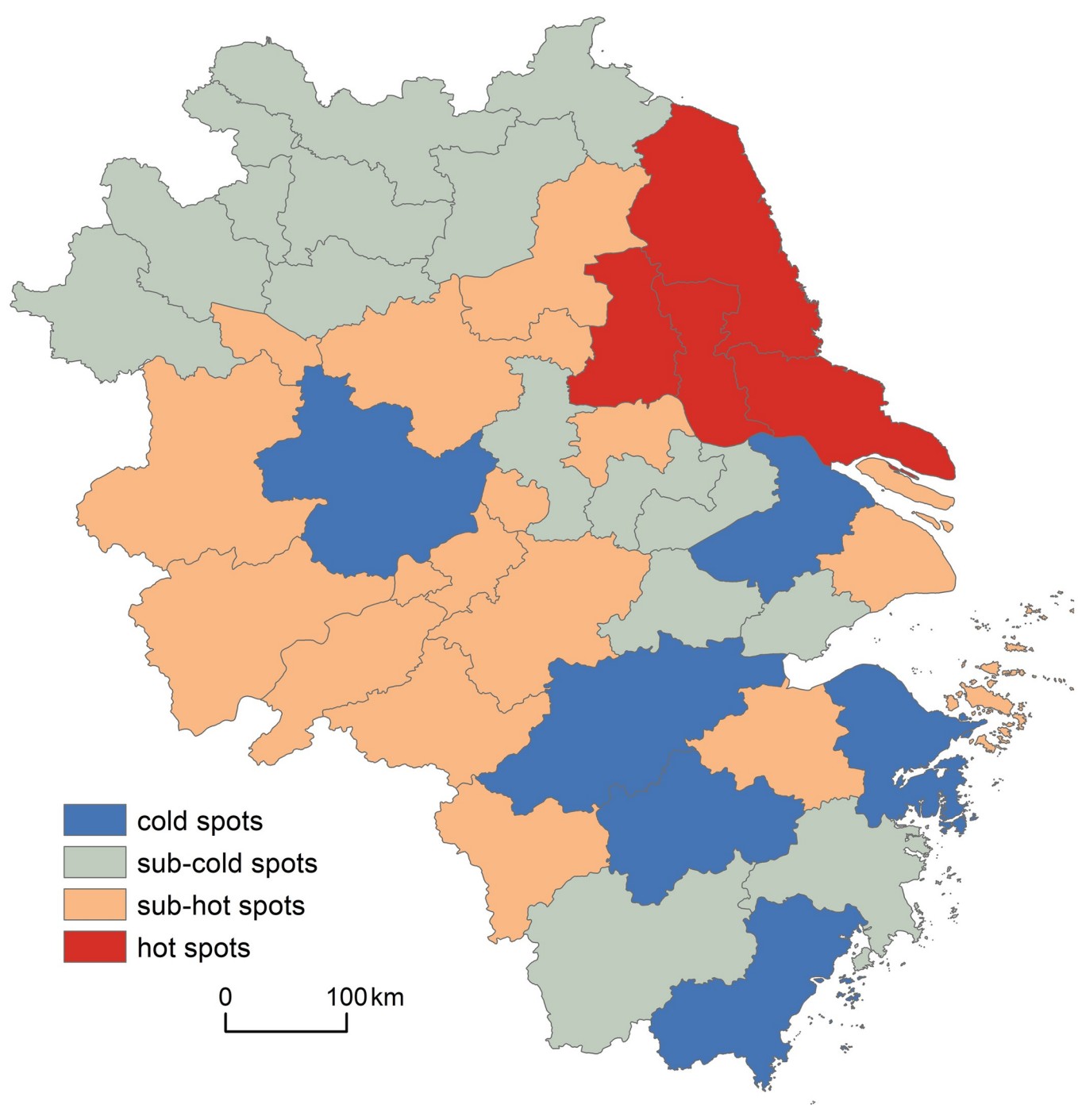

**Fig 3. Analysis of cold and hot spots of aging rate in the Yangtze River Delta in 2020.**

Most cities have a low density of aging populations, and more than half (51.22%) have an older adults agglomeration degree of 3–6, such as 3.75 for Huai'an. Second largest is the sparse area with 8 cities, accounting for 19.51% and mainly distributed in the western inland area, among which Lishui has the lowest concentration at 1.12. It is followed by the medium-density area with 6 cities (14.63%), mainly concentrated in Jiangsu Province (such as Nantong, Taizhou, Yangzhou). The high-density area represents 12.2% of the Yangtze River Delta's cities

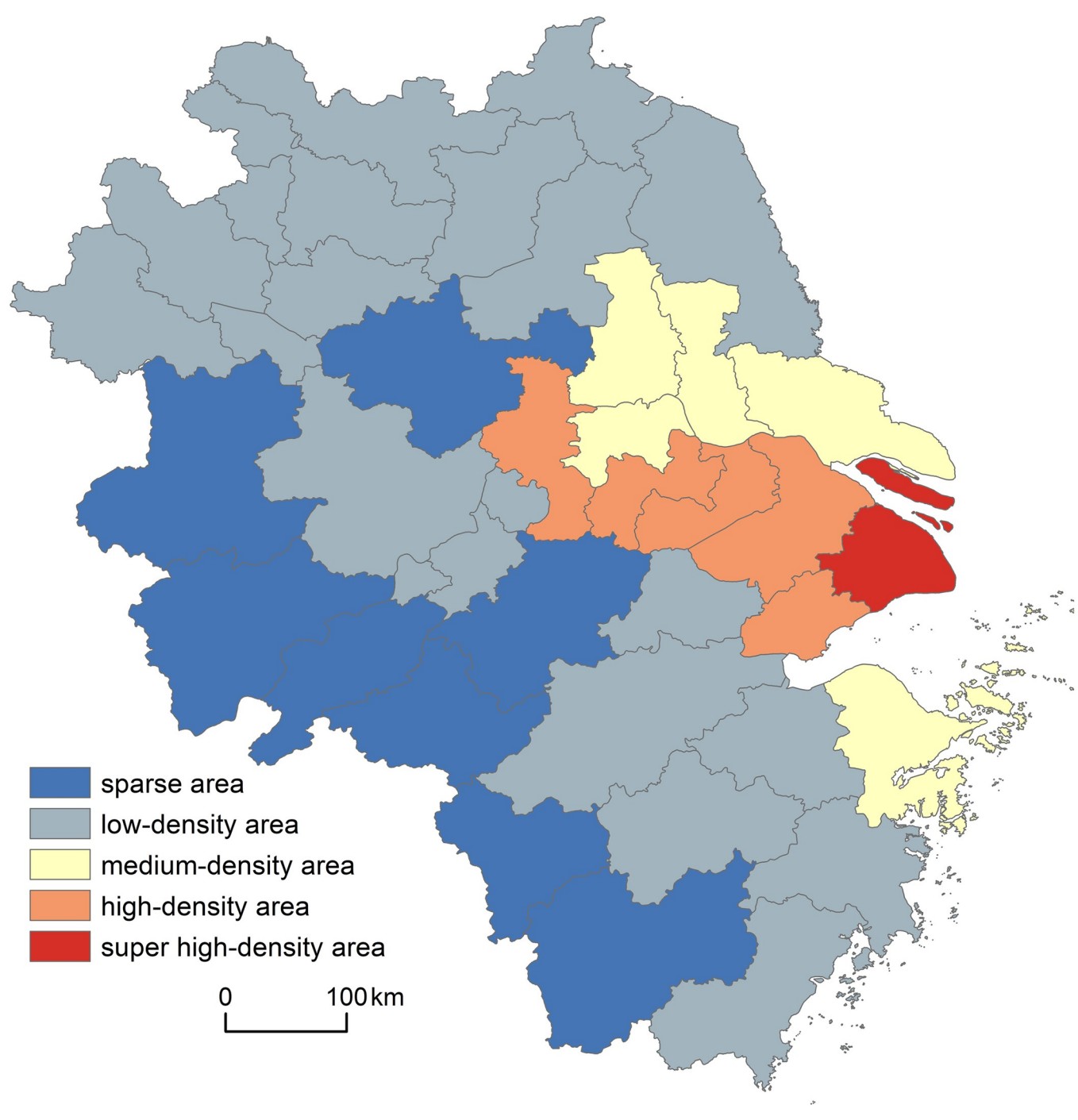

**Fig 4. Agglomeration degree of older adults in the Yangtze River Delta in 2020.**

and includes Suzhou, Wuxi, Nanjing, and other locations along the Beijing-Shanghai railway. Finally, the older adults' density of Shanghai is 32.15, much higher than that of neighboring cities, indicating that it is a super-high-density area. Overall, Shanghai has a much higher concentration of older adults than other cities, indicating a much higher demand for elderly care. Consequently, preference should be given to Shanghai for the allocation of elderly care resources.

### The spatial distribution of elderly care resources in the Yangtze River Delta

Spatial cluster analysis and natural break classification were employed to classify and characterize the spatial pattern of the allocation of elderly care resources in the Yangtze River Delta in 2020, i.e., life services, ecological environment, social security, and medical and health resources (see Fig 5).

(1) There are significant spatial differences in life service resources. The supply of elderly care facilities in places such as Nantong, Taizhou is severely insufficient, whereas some cities such as Huzhou have the most life service resources, and Shanghai belongs to the intermediate type.

As Fig 5A illustrates, if measured by the number of elderly care facilities per 10,000 older adults ($Y_1$), life service resources in the Yangtze River Delta can be divided into five categories. The first category has the lowest number of elderly care facilities ($0.527 < Y_1 \leq 0.667$), with 11 cities (26.83% of the total), mainly distributed in cities such as Yancheng and Wuhu. The

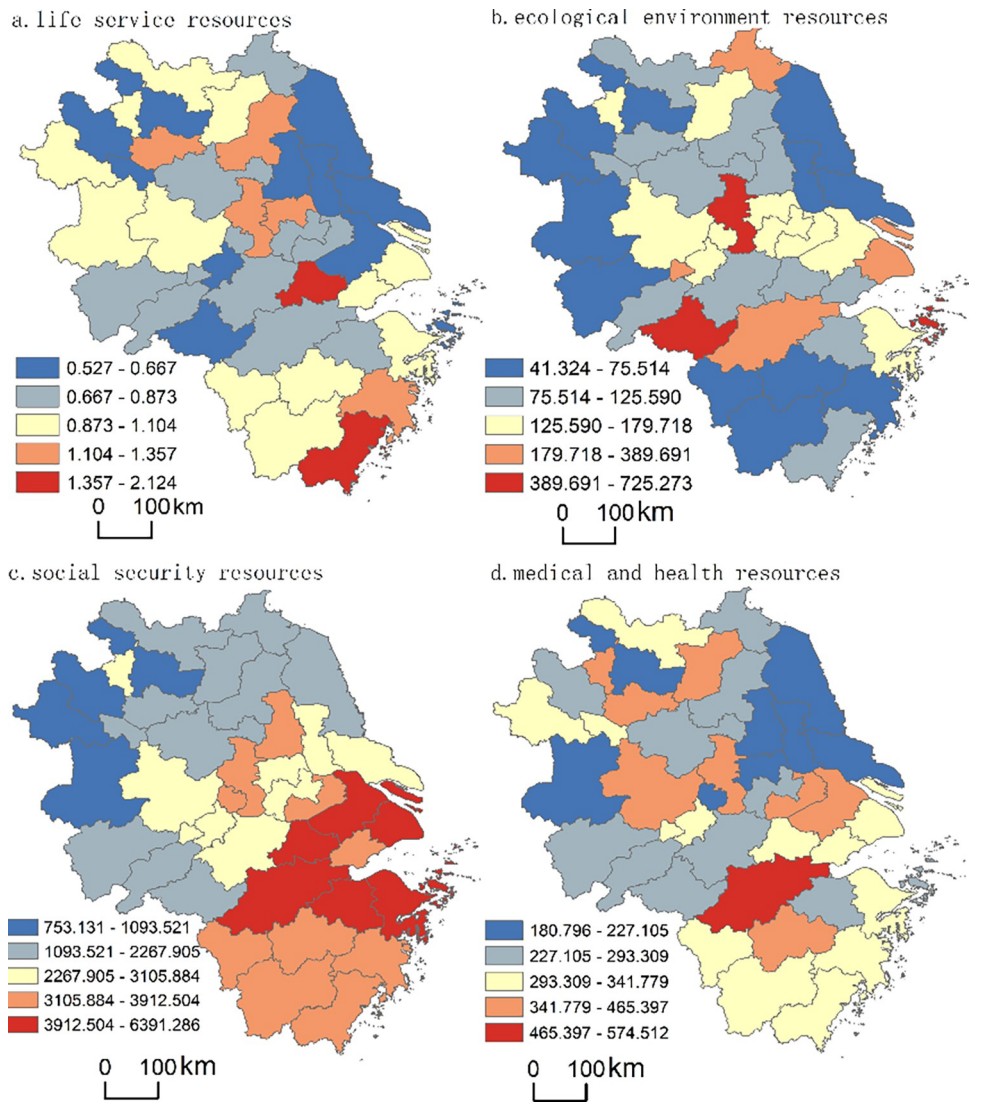

**Fig 5. Spatial cluster analysis of elderly care resources in the Yangtze River Delta in 2020.**

second category refers to areas with $0.667 < Y_1 \leq 0.873$, again 11 cities (26.83%), and mainly distributed in areas such as Wuxi and Changzhou. The third category, $0.873 < Y_1 \leq 1.104$, represents 12 cities (29.27%), e.g., Lu'an and Shanghai. The fourth category, $1.104 < Y_1 \leq 1.357$, contains 5 cities (12.2%), mainly located in cities such as Zhenjiang and Nanjing. The final category consists of Huzhou and Wenzhou (4.87% of the total) with the largest number of elderly care facilities ($1.357 < Y_1 \leq 2.124$). The results clearly indicate large spatial differences in life service resources. There is a serious shortage of elderly care facilities in cities such as Nantong and Taizhou, whereas some cities such as Huzhou have the most life service resources. Shanghai's life service resources are of an intermediate type.

(2) There are significant spatial differences in ecological environment resources. The supply of greenspace resources in cities such as Nantong and Yancheng is insufficient, while that in cities such as Huangshan, Zhoushan, and Nanjing is the highest.

As Fig 5B illustrates, the area of park green spaces per 10,000 older adults ($Y_2$) is used to measure ecological environment resources in the Yangtze River Delta, which can be classified into five categories. The first category is $41.324 < Y_2 \leq 75.514$ (the lowest level), and comprises 12 cities (29.27%), mainly distributed in cities such as Lishui and Quzhou. The second category, $75.514 < Y_2 \leq 125.590$, also comprises 12 cities (29.27%), such as Bengbu, Wenzhou. There are 10 cities (24.39%), such as Suzhou and Wuxi, in the third category ($125.590 < Y_2 \leq 179.718$); the fourth category ($179.718 < Y_2 \leq 389.691$) comprises 4 cities (9.76%), e.g., Shanghai, Lianyungang. The fifth category, $389.691 < Y_2 \leq 725.273$ (the highest level), comprises 3 cities such as Nanjing (7.32% of the total). The results demonstrate enormous spatial differences in ecological environment resources in the Yangtze River Delta (the range is 683.95) with a particularly serious lack of ecological and environmental resources in places such as Nantong, Yancheng, and Jinhua, while the most abundant resources are in cities such as Huangshan, Zhoushan, and Nanjing.

(3) There are significant spatial differences in social security resources. The supply of these resources in less-developed areas such as Lu'an is seriously insufficient, while that in developed locations such as Shanghai is relatively the highest; cities such as Nantong and Taizhou belong to the intermediate type.

As Fig 5C illustrates, social security resources are measured by the number of urban basic old-age insurance participants per 10,000 older adults ($Y_3$) and can be divided into five categories. The first category is $753.131 < Y_3 \leq 1093.521$ (the lowest), comprising 4 cities (9.76% of the total), including Suzhou and Lu'an. The second category, $1093.521 < Y_3 \leq 2267.905$, is composed of 11 cities (26.83%) mainly distributed in Chuzhou, Huangshan, etc. The third category, $2267.905 < Y_3 \leq 3105.884$, comprises 9 cities (21.95%) such as Nantong, Wuhu. The fourth category ($3105.884 < Y_3 \leq 3912.504$) contains 10 cities (24.39%), such as Quzhou and Jinhua. The fifth and final category was $3912.504 < Y_3 \leq 6391.286$ (the highest level), with 7 areas (17.07%), including Shanghai and Suzhou. From this perspective, social security resources vary significantly spatially. Underdeveloped cities such as Fuyang, Bozhou, and Suzhou have serious shortages of social security resources, while developed cities such as Shanghai and Hangzhou have the most abundant, with Nantong, Taizhou and some other cities being of an intermediate type ($2267.905 < Y_3 \leq 3105.884$).

(4) There are significant spatial differences between medical and health resources. The supply of medical and health resources in Nantong is insufficient, while that in Hangzhou is the highest, and Shanghai belongs to the intermediate type.

As Fig 5D illustrates, medical and health resources are measured by the number of beds per 10,000 older adults ($Y_4$), which can be divided into five categories. There are 8 cities with $180.796 < Y_4 \leq 227.105$ categorised as the first type, including Lu'an and Nantong. The second category, $227.105 < Y_4 \leq 293.309$, contains 11 cities (26.83%), e.g., Lu'an and Huangshan. The

third category (293.309<$Y_4$≤341.779) comprises 13 cities (31.71%), such as Wenzhou, Shanghai. There are 8 cities (19.51%) belonging to the fourth category (341.779<$Y_4$≤465.397) and are located in Nanjing, Jinhua, and some other areas. The final category represents the amplest medical and health resources with a range of 465.397<$Y_4$≤574.512 and comprises only Hangzhou, 2.44% of the total. Thus, there are large spatial differences in medical and health resources, and the supply in Nantong, Taizhou, Yancheng, etc. is inadequate. Generally speaking, medical and health resources in places such as Hangzhou are the most abundant, whereas those in Shanghai belong to the middle type. Despite the fact that the area is developed, Shanghai's high degree of aging results in limited medical and health resources for its older people, which should be the government's focus.

## Coupling and matching relationship between elderly care resources and older adults in the Yangtze River Delta

### Comprehensive evaluation of the coupling relationship

(1) In the Yangtze River Delta, the overall coupling effect between the older adults and the elderly care resources is strong, and more than half of the cities have a relatively high correlation.

According to the above indicator systems and Eqs (4)–(8), the total coupling degree between the older adults and the elderly service resource in the Yangtze River Delta in 2020 is 0.6661, which belongs to a relatively high correlation, indicating that the overall coupling effect between the two systems is strong. (For comparison, the coupling data in this study are uniformly reserved for 4 decimal places).

This study further calculated the coupling degree between the older adults and the elderly care resources in each city in 2020, comprehensively evaluated their relationship, and used ArcGIS to perform a cluster analysis of the coupling degree in combination with classification standards of the coupling correlation, as Fig 6 shows. The overall coupling relationship between the older adults and elderly care resources in the Yangtze River Delta can be divided into three types, as follows:

The first type is moderate correlation, including 16 cities (accounting for 39.02% of all cities), mainly distributed in the east and west, such as Jinhua, Taizhou, Nantong; the second type is relatively high correlation, including 23 cities (56.10%), widely distributed in the southeast, south and north, such as Shanghai, Suzhou, Suzhou; the third type is high correlation, including only Xuzhou and Lianyungang (4.88%). It can be seen that the elderly care resources in more than half of the cities in the Yangtze River Delta have a relatively high correlated with the older adults, followed by a moderate correlation, and very few areas are highly correlated.

(2) Social security resources have the strongest coupling correlation with the older adults while the park green space resources have the weakest one. In brief, the overall coupling correlation between the elderly care resources and the older adults in the Yangtze River Delta in 2020 is relatively high, indicating a strong coupling effect. More specifically, social security resources have the greatest coupling effect on the older adults (coupling degree is 0.6672), followed by medical beds, elderly care facilities, and park green space resources in turn.

### Life service resources and the older adults show mainly moderate and high coupling

Based on the coupling coordination model and Eq (9), the coupling coordination degree and spatial matching between every kind of elderly care resources and older adults in each city of the Yangtze River Delta were calculated, as Fig 7 reveals.

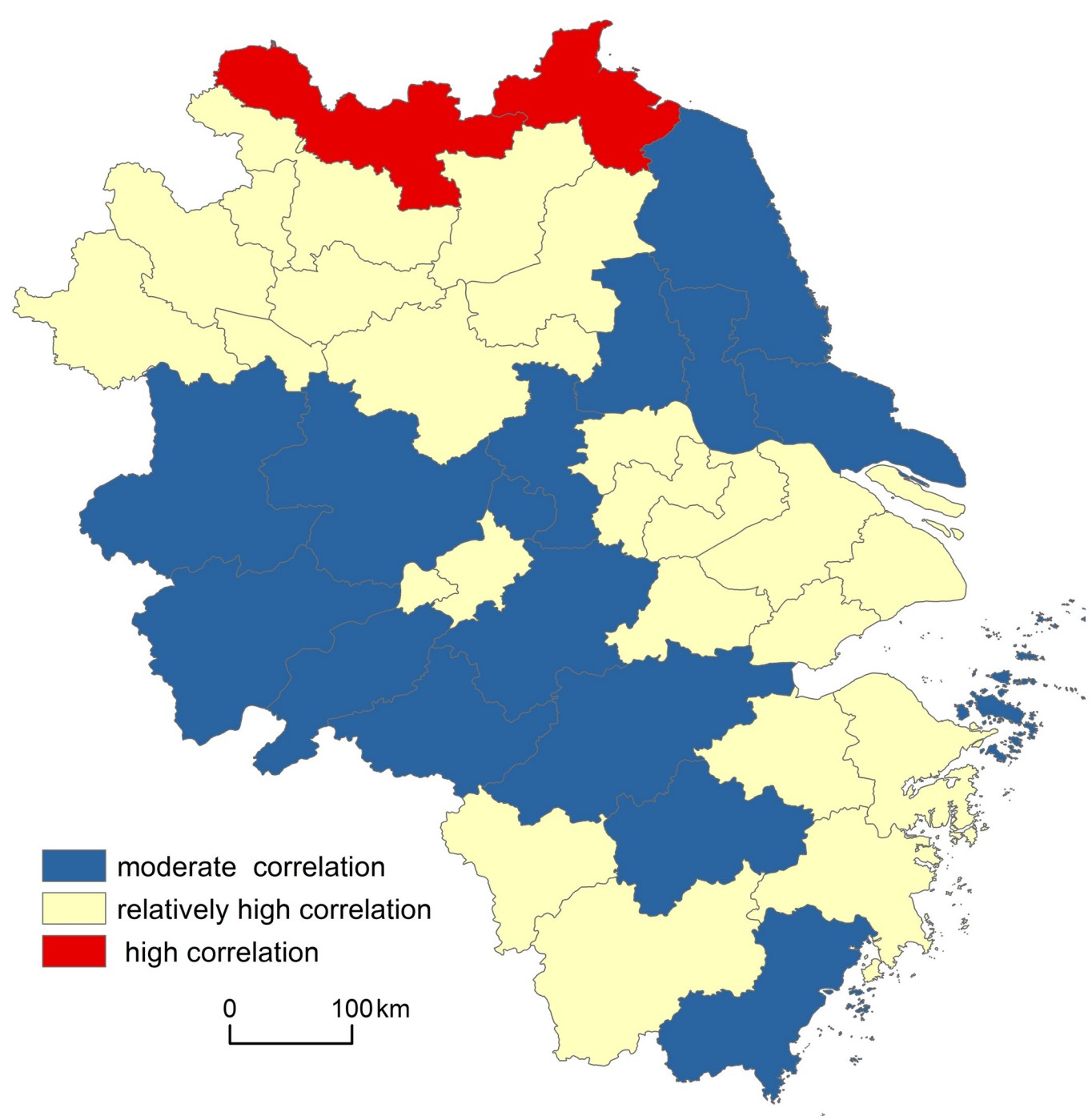

**Fig 6. Comprehensive evaluation of the coupling relationship between elderly care resources and older adults in the Yangtze River Delta in 2020.**

Furthermore, natural break classification was adopted to classify the matching degree of each kind of elderly care resources and older adults into five categories: severe mismatch (highly general mismatch), general mismatch (no coupling), barely matched (low coupling), intermediate match (moderate coupling), and high match (high coupling), as Fig 8 indicates.

Fig 8A illustrates that a severe mismatch between life service resources and older adults occurs in 8 cities (19.51%), including Yangzhou and Jinhua. In 4 cities (9.76%), such as Hefei

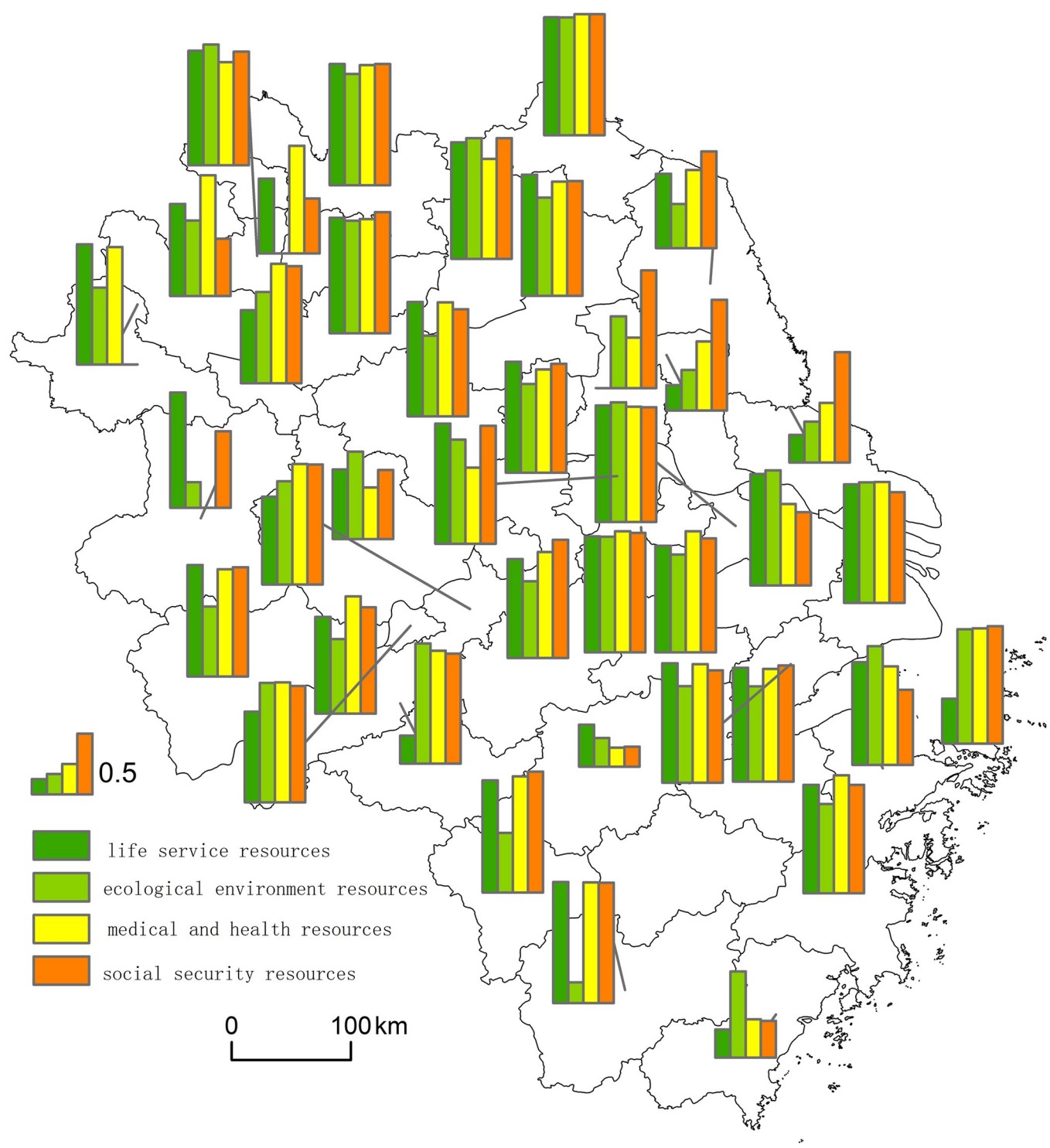

**Fig 7. Spatial matching of elderly care resources and older adults in the Yangtze River Delta.**

and Suzhou, life service resources and the older adults are generally mismatched, and in 7 cities (17.07%), including Wuhu and Ningbo, they barely match. Additionally, 11 cities (26.83%), including Nanjing and Huzhou, have an intermediate match. The remaining 11 cities

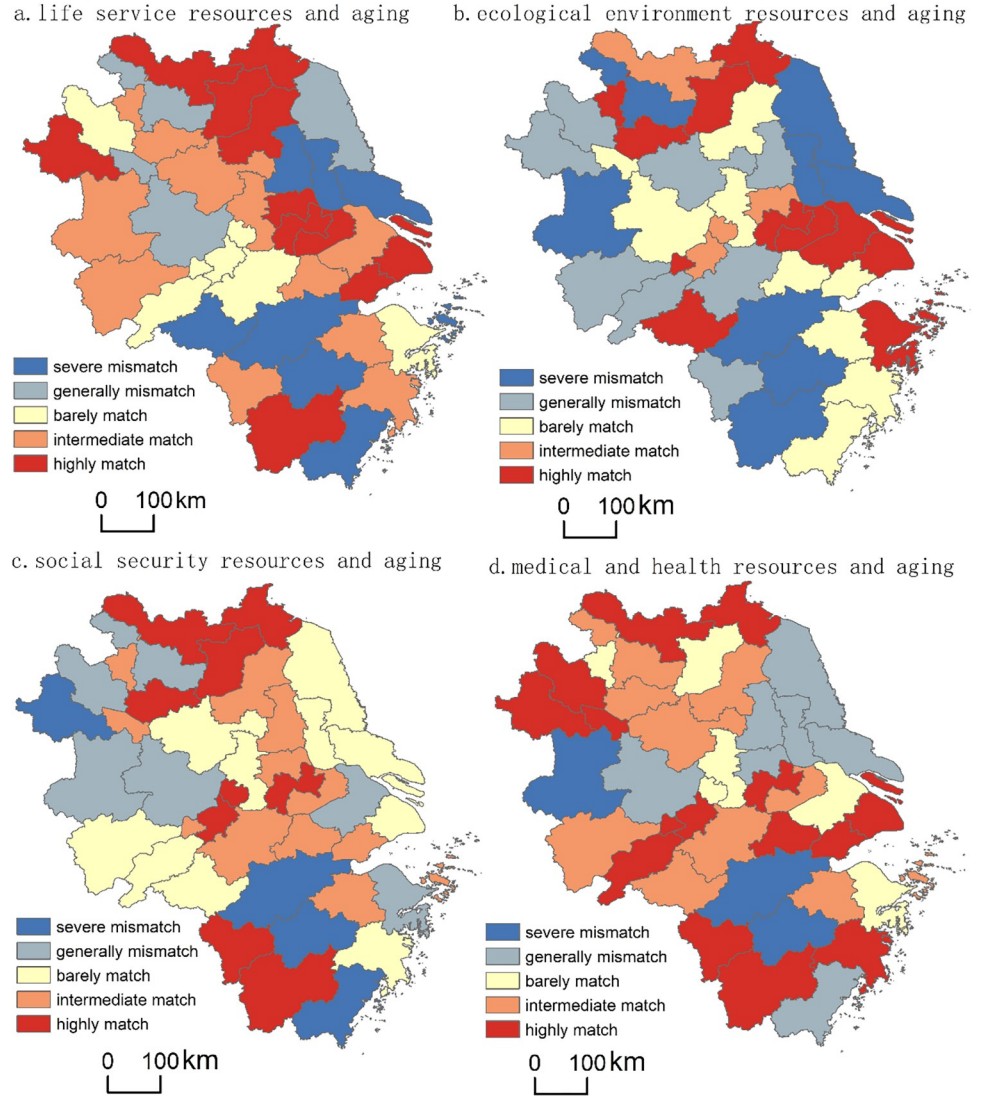

**Fig 8. Spatial matching types of elderly care resources and aging in the Yangtze River Delta.**

(26.83%), including Changzhou and Shanghai, show that life service resources and the older adults are highly matched. Overall, life service resources and older adults in the Yangtze River Delta see predominantly medium and high matches. However, these resources in some cities (e.g., Nantong and Taizhou) are seriously mismatched with the older adults.

## Ecological environment resources and the older adults show mainly high coupling

In 8 cities (19.51%) in the Yangtze River Delta, such as Jinhua and Taizhou, the ecological environment resources and the older adults are severely mismatched, whereas a general mismatch appears in 8 cities (19.51%) such as Quzhou, Fuyang. Another 9 cities (including Hefei and Huzhou, 17.07%) are barely matched with the older adults, and another 4 cities (9.76%), including Zhenjiang and Ma'anshan, are moderately matched. The high match type is the most common, with 12 cities (29.27%) including Zhoushan and Shanghai (Fig 8B). It indicates

that ecological environment resources in the Yangtze River Delta are mainly of a high couple with the older adults (such as Suzhou and Shanghai), but this is not the case in Nantong, Taizhou among others where there is a serious mismatch with the older adults.

## Social security resources and the older adults show mainly moderate coupling

In the Yangtze River Delta, there are 4 cities (9.76%), e.g., Jinhua and Hangzhou, demonstrating a severe mismatch of social security resources and the older adults, while there are 6 cities (14.63%) showing a general mismatch, such as in Suzhou (Fig 8C).

The barely matched category contains 10 cities (24.39%, including Shanghai and Nantong) whereas another 12 (29.27%), such as Jiaxing and Yangzhou, are moderately matched. The final category, a high degree of coupling, contains 9 cities (21.95%), including Changzhou and Bengbu. The results show that, for the most part, social security resources in the Yangtze River Delta are matched with the older adults at an intermediate level. Social security resources in cities such as Hangzhou, however, are seriously mismatched with the older adults, and Shanghai and Taizhou are examples of areas where social security resources barely match the older adults. In others, such as Xuzhou and Bengbu, the coupling is at the highest level in the Delta region.

## Medical and health resources and the older adults show mainly moderate coupling

Fig 8D illustrates 3 cities (including Hangzhou) in the Yangtze River Delta (7.32%) where medical and health resources are seriously mismatched with the older adults. In addition, 7 cities (17.07%), including Nantong and Taizhou, reveal a general mismatch between medical and health resources and the older adults, while another 6 cities (14.63%), including Suzhou and Ningbo, show them barely matching. The final two categories are intermediately matched, such as Huangshan and Zhoushan (10 cities, 24.39%), and highly matched (15 cities, 36.59%), as in the case of Shanghai and Huzhou. Overall, medical and health resources in the Yangtze River Delta mainly have a high coupling degree with the older adults; however, in areas such as Jinhua and Hangzhou, they are seriously mismatched.

## Conclusions and discussion

Based on previous research, this study proposes the connotation and classification of elderly care resources from broad and narrow perspectives. We elaborated the narrow sense of elderly care resources, namely all material and tangible resources available for the social elderly care, divided into four types: life service resources (such as institution and community elderly care service facilities), ecological environment resources (such as the park green space and air temperature), social security resources (such as pension insurance), and medical and health resources (such as the number of medical beds and medical staff available). The main conclusions are summarized as follows.

In relation to the demand for elderly care, the aging population in the Yangtze River Delta is mainly at the moderate aging stage, followed by the primary stage. The central Jiangsu region (including Nantong and Taizhou), which is in the severe stage, needs elderly care resources. Population-aging hot spots (such as Nantong) have a greater demand for elderly care, whereas those in cold spots (such as Hefei) have relatively smaller demand levels. The older adults concentration and care needs in Shanghai are much higher than in other regions, while provincial capitals (Nanjing, Hangzhou, and Hefei) are all in the primary stage.

Specifically, except for Nanjing, which has a high density of older people, the remaining provincial capitals have a low-density distribution.

Concerning the supply of elderly care, the spatial distribution of resources varies distinctly among the cities in the Yangtze River Delta. Huzhou, which is close to Shanghai but has a low-density older adults, has the most abundant life service resources, whereas in central Jiangsu (such as in Nantong and Taizhou) they are insufficient. Tourist hot spots (such as Huangshan, Zhoushan, and Nanjing) are rich in ecological and environmental resources, while cities such as Nantong and Yancheng are inadequately supplied. Shanghai has the most abundant social security resources, whereas economically underdeveloped areas (such as Lu'an) have serious shortages. Finally, Hangzhou has the most medical and health resources, whereas cities such as Nantong are severely undersupplied.

Regarding the coupled coordination of supply and demand, there is a strong overall coupling between the older adults and elderly care resources in the Yangtze River Delta. Among them, the coupling of life service resources and the older adults are mainly medium or high (intermediate and high adaptability). For example, the life service resources in Huzhou and Nanjing are moderately coupled with the older adults, and those in Changzhou are highly coupled. However, those in cities such as Nantong and Taizhou are seriously mismatched. Ecological environment resources and the older adults are mainly highly coupled (such as in Suzhou and Changzhou), but these kinds of resources in places such as Nantong, Taizhou, Jinhua, and Hangzhou are seriously mismatched; the older adults and those in Huzhou, Nanjing, etc. are barely matched. In addition, the coupling of social security resources and the older adults are at the intermediate level (distributed in cities such as Wuxi and Huzhou). However, there is a serious mismatch in cities such as Hangzhou, while social security resources in Shanghai, Nanjing, Nantong, and Taizhou and some others barely match the older adults. Those in cities such as Changzhou, Xuzhou, and Bengbu, however, are highly matched. Finally, medical and health resources are overall highly coupled with the older adults (e.g., Huzhou and Jiaxing). However, in cities, e.g., Jinhua, Lu'an, and Hangzhou, they are seriously mismatched and in cities such as Nantong and Taizhou, they are generally mismatched. They are barely matched in cities (such as Suzhou) and moderately matched in Huangshan, Zhoushan, etc. Overall, Nantong, which has the highest degree of aging, has a serious mismatch in both life service resources and ecological environment resources, while provincial capitals, with a low degree of aging and a low population density of older adults, have a mismatch in social security, medical, and healthcare resources. Medical and health resources in underdeveloped areas such as Lu'an are severely mismatched with the older adults. Although social security resources in Shanghai are the most abundant, they are barely matched with the older adults because the concentration of the older adults in Shanghai is much higher than in other regions.

Accordingly, a path for optimizing the spatial allocation of older adults and elderly care services is proposed. First, the top-level design of the optimal allocation of elderly care resources in the Yangtze River Delta region should be strengthened, including improving the system and reinforcing organizational and personnel guarantees of the optimal allocation of those resources.

Second, based on the integration strategy of the Yangtze River Delta, strengthen cooperation in elderly care services in the Yangtze River Delta, integrate elderly care resources in the regions within the Yangtze River Delta, adjust the spatial allocation of elderly care resources, and promote the integrated development of elderly care in the Yangtze River Delta. For regions where the elderly care resources do not match the older adults, given the actual demands of the older adults and the spatial agglomeration characteristics, the government should rationally adjust the direction of the supply side of elderly care resources and increase

the supply of relevant resources in areas where life services and ecological environment resources are seriously mismatched (e.g., Nantong).

Moreover, it is important to expand the coverage of social security resources in Lu'an and other underdeveloped areas and strengthen the rational allocation of medical and health resources, on which older adults rely, in severely mismatched areas to promote the balanced and coupled development of elderly care resources. For the Yangtze River Delta, measures should be taken to ensure that Shanghai plays a leading role in high-tech and high medical level, accelerate the implementation of the Yangtze River Delta integration strategy, strengthen regional connections, and promote the integrated development of smart elderly care in the Yangtze River Delta. Accelerate the implementation of the Yangtze River Delta integration strategy, promote the integration of transportation in the Yangtze River Delta (such as the passage of Shanghai and Suzhou subways), and the integration of medical care in the Yangtze River Delta (such as the opening of remote medical insurance), providing feasibility for the integration of elderly care in the Yangtze River Delta.

Finally, it is necessary to reasonably guide older adults in Shanghai to move to nearby areas with rich elderly care resources and low aging levels for elderly care. On the one hand, although Shanghai, as an international metropolis, has the most abundant social security resources, middle-level life services, ecological environment, and medical resources, its concentration of older adults is far higher than in other areas. Consequently, social security resources are barely sufficient for the older adults, exerting enormous pressure to support those older adults. This issue presents a major challenge to the economic and social development of all supercities.

On the other hand, primary aging areas (e.g. the southeast coast of Zhejiang), places adjacent to Shanghai with a lower degree of aging (such as Huzhou with rich life resources and Suzhou with rich social security resources), and provincial capital cities (such as Hangzhou with sufficient medical and health care and social security resources) are available to support older adults from cities such as Shanghai, which have serious aging problems. It would not only alleviate care pressure and resource shortages in outflow areas but also make full use of the characteristic resources of inflow areas and develop characteristic industries. Furthermore, it could facilitate the integrated development of the Yangtze River Delta, provide scientific evidence to help break down administrative divisions in China and solve the problem of elderly care in supercities through regional linkage integration.

Given the difficulty in obtaining some of the latest city-scale data in the Yangtze River Delta, elderly care resources have not been analyzed from a broad perspective. Furthermore, the challenge of data accessibility and comparability of four types of elderly care resources led us to choose only one indicator to represent each resource. There may be a potential limitation in adequately characterizing four kinds of elderly care resources, so future research needs to consider increasing types of mental elderly care resources to improve the classification of elderly care resources, such as spiritual and cultural resources (e.g., service workers per 10,000 older adults living in cities in the Yangtze River Delta in 2020), and incorporate more indicators for each elderly care resource, so as to depict the status of various elderly care resources as comprehensively and deeply as possible.

## Author Contributions

**Conceptualization:** Lianxia Wu.

**Data curation:** Weihua Guan.

**Formal analysis:** Lianxia Wu, Weihua Guan.

**Investigation:** Zuyu Huang.

**Methodology:** Lianxia Wu, Linyi Qian.

**Project administration:** Lianxia Wu.

**Software:** Linyi Qian.

**Supervision:** Lianxia Wu, Zuyu Huang.

**Writing – original draft:** Lianxia Wu.

**Writing – review & editing:** Yinhuan Li.

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
