## [Decision Letter · Decision Letter 0]

25 Aug 2023

PONE-D-23-22426Spatial Coupling Relationship between Older Adults and Elderly Care Resources in the Yangtze River DeltaPLOS ONE

Dear Dr. Li,

Thank you for submitting your manuscript to PLOS ONE. After careful consideration, we feel that it has merit but does not fully meet PLOS ONE’s publication criteria as it currently stands. Therefore, we invite you to submit a revised version of the manuscript that addresses the points raised during the review process.

We look forward to receiving your revised manuscript.

Kind regards,

Changjian Wang

Academic Editor

PLOS ONE

Journal Requirements:

"National Social Science Fund in China (LXW,19CRK010), the Humanity and Social Sciences Foundation of Ministry of Education of China (LXW, 18YJC840043; ZHP,19YJAZH023; WHG,19YJC840032), Fundamental Research Funds for the Central Universities (LXW,2020ECNU-HLYT048; LYQ,2022QKT001), National Natural Science Foundation (ZHP, 42001161), Nanjing Social Science Foundation Project (WHG, 20ZX01)."

4. We note that [Figures 1-7] in your submission contain [map/satellite] images which may be copyrighted. All PLOS content is published under the Creative Commons Attribution License (CC BY 4.0), which means that the manuscript, images, and Supporting Information files will be freely available online, and any third party is permitted to access, download, copy, distribute, and use these materials in any way, even commercially, with proper attribution. For these reasons, we cannot publish previously copyrighted maps or satellite images created using proprietary data, such as Google software (Google Maps, Street View, and Earth). For more information, see our copyright guidelines: http://journals.plos.org/plosone/s/licenses-and-copyright.

a. You may seek permission from the original copyright holder of Figures 1-7 to publish the content specifically under the CC BY 4.0 license.  

Reviewers' comments:

Reviewer's Responses to Questions

**Comments to the Author**

1. Is the manuscript technically sound, and do the data support the conclusions?

Reviewer #1: Yes

Reviewer #2: Yes

2. Has the statistical analysis been performed appropriately and rigorously? 

Reviewer #1: N/A

Reviewer #2: Yes

3. Have the authors made all data underlying the findings in their manuscript fully available?

Reviewer #1: No

Reviewer #2: Yes

4. Is the manuscript presented in an intelligible fashion and written in standard English?

Reviewer #1: Yes

Reviewer #2: Yes

5. Review Comments to the Author

Reviewer #1: This study explored the spatial coupling relationship between the older adults and elderly care resources in the Yangtze River Delta. I think this manuscript needs substantial revisions before it is considered for publication in PLOS ONE. Here are my comments:

1. The language of this paper needs to be further polished and improved.

2. In the section of Introduction, the current situation of population ageing in the Yangtze River Delta should be further illustrated in details.

3. The structure of the paper needs to be improved, especially the Literature Review part. For example, the authors should list the key ideas in the section of literature review. In addition, the logic of some contents in the section of literature review is confusing, such as the second and the third paragraph of the literature review.

4. It would be better to include a map to show the research area.

5. The authors should check their manuscript more carefully before submitting the manuscript. There are some obvious flaws in this manuscript, such as Line 190 and Line 248.

6. Table 1: It would be better if the authors define the scope of different facility resources in more details. For example, what types of infrastructure facilities do the “elderly care facilities” contain? In addition, can the indicator of “number of medical beds per 10,000 older adults” fully represent the medical and health resources needed by older adults?

7. In the manuscript, the terms of “elderly care resources”, “elderly care service resources” and “elderly care facilities resources” are very confusing to the reader.

8. Line 419: What is “life service resources”? There is a mismatch between the text and Table 1 in terms of the names of indicators.

9. In the section of conclusion, the authors should list more targeted policy implications based on the analysis results. In addition, some implications seem to be unrealistic, such as Line 534: it is necessary to reasonably guide older adults to move to areas with rich elderly care resources and low aging levels.

10. The authors need to elaborate more on the contribution of this paper to our knowledge. Given the limited available spaces for older adults’ daily activities, does it make sense to carry out analysis at such a large spatial scale?

Reviewer #2: I would like to pay my thanks to you for entrusting me with this opportunity to review an article for your journal. After reading this article critically, I think this article is very clear and well written. The Yangtze River Delta is an important economic zone in China and also one of the regions with severe population aging issues. The relationship between elderly care resources and older adults is crucial for China's strategy to address population aging. Therefore, the study is pretty important in the context of China. The authors put their best into conducting this research. The abstract, introduction, literature and results are well written and explained.

Though, I have a few suggestions which are expected to be considered in the revision of the manuscript and future research.

(1) Third-level indicators of the coupling system need to be chosen more carefully and need to be elaborated.

(2) In the "Literature Review", it was mentioned that the national strategy of "Yangtze River Delta Integration" is particularly important for promoting the integration of elderly care resources. However, the importance of "Yangtze River Delta Integration" in promoting the integration of elderly care resources and how to use the Yangtze River Delta Integration Strategy to promote the integration of elderly care resources have not been clearly reflected in the countermeasures. It is necessary to highlight or supplement the suggestions in the countermeasures.

(3) At present, there is a lack of literature in the reference list that has been submitted to the journal. Therefore, it is recommended to appropriately add references to the relevant topics that have been published in the PLOS ONE.

6. PLOS authors have the option to publish the peer review history of their article (what does this mean?). If published, this will include your full peer review and any attached files.

Reviewer #1: No

Reviewer #2: No

---

## [Author Response · Author response to Decision Letter 0]

3 Oct 2023

Dear Reviewers:

Thank you for your careful reading of our manuscript and valuable comments, which helps us significantly improve the revised version of the paper. We believe that we have carefully addressed these comments in the revised version. The details of our responses are shown in file 'Responses to Reviewers', and the texts of the revised manuscript based on the comments are remarked and tracked in file 'Revised Manuscript with Tracked Changes'. For the 'Revised Manuscript with Tracked Changes', to hide the formatting revision and show our insertions and deletions, please choose "Review"-" Show Markup"- no "formatting". 

Best,

Yinhuan Li

---

## [Editor Report · Decision Letter 1]

24 Oct 2023

Spatial Coupling Relationship between Older Adults and Elderly Care Resources in the Yangtze River Delta

PONE-D-23-22426R1

Dear Dr. Li,

We’re pleased to inform you that your manuscript has been judged scientifically suitable for publication and will be formally accepted for publication once it meets all outstanding technical requirements.

Kind regards,

Changjian Wang

Academic Editor

PLOS ONE

---

## [Editor Report · Acceptance letter]

30 Oct 2023

PONE-D-23-22426R1 

Spatial Coupling Relationship between Older Adults and Elderly Care Resources in the Yangtze River Delta 

Dear Dr. Li:

I'm pleased to inform you that your manuscript has been deemed suitable for publication in PLOS ONE. Congratulations! Your manuscript is now with our production department. 

Kind regards, 

on behalf of

Prof. Dr. Changjian Wang 

Academic Editor

PLOS ONE